

# How 3d volcanic stratigraphy constrains headscarp collapse scenarios : the Samperre Cliff case study (Martinique Island, Lesser Antilles)

Marc Peruzzetto[1], Yoann Legendre[1], Aude Nachbaur[2], Thomas J.B. Dewez[1], Yannick Thiery[1], Clara Levy[1], and Benoit Vittecoq[2]

[1]BRGM, F-45060 Orléans, France
[2]BRGM Martinique, 97200 Fort-de-France, Martinique

**Correspondence:** Marc PERUZZETTO (m.peruzzetto@brgm.fr)

**Abstract.** Gravitational instabilities can be significant threats to populations and infrastructures. For hazard assessment, it is of prior importance to estimate the geometry and volume of potential unstable masses. This characterization can be particularly difficult in volcanic context due to the succession of deposition and erosion phases. Indeed, it results in complex layering geometries where the interfaces between geological layers may be neither parallel nor planar. Geometry characterization is all

the more complex when unstable masses are located in steep and hardly accessible landscapes, which limits data acquisition. In this work, we show how remote observations can be used to estimate the surface envelope of an unstable mass on a volcanic cliff. We use ortho-photographs, aerial views and topographic surveys to (i) describe the different geological units of the cliff, (ii) identify stable and unstable units, (iii) infer the paleo-morphology of the site and (iv) estimate potential unstable volumes. We use the Samperre cliff in Martinique (Lesser Antilles, French West Indies) as a study site, where recurrent destabilizations

since at least 1988 have produced debris flows that threaten populations and infrastructures. Our analysis suggests that the destabilizations occurring on the cliff may be associated to the re opening of a paleo-valley filled by pyroclastic materials. We estimate that between $3.5 \times 10^6$ m$^3$ and $8.3 \times 10^6$ m$^3$ could still be mobilized by future destabilizations in the coming decades.

## 1  Introduction

Volcanoes alternate construction phases, when eruption materials widen and heighten the volcanic edifice, and destruction

phases leading to the progressive dismantling of the edifice (Thouret, 1999). Such destruction phases occur sometimes very violently during caldera forming eruptions, e.g. during the Samalas 1257 CE eruption (Lavigne et al., 2013) or the Tambora 1815 CE eruption (Stothers, 1984). The erosion of volcanic edifices also results from gravitational insatiabilities because of the mere construction mechanism of volcanoes, by progressive accumulation of eruptive materials (Ramalho et al., 2013). In volcanic context, gravitational instabilities are favoured or triggered by various factors (see Schaefer et al. (2019) for a review).

They include steep slopes (e.g. on steep walled andesitic edifices, Voight, 2000), hydrothermal alteration (Revil et al., 2020; Heap et al., 2021), hydrothermal pressurization (Reid, 2004), rainfall (van Wyk de Vries et al., 2000) and earthquakes (Coviello et al., 2021). Eruptions favor gravitational instabilities (Siebert et al., 1987; Durand et al., 2018), but are not necessarily the





main triggering factor (Cutler et al., 2022). The scale of instabilities varies from a few m³ (e.g. for rockfalls, Durand et al., 2018) to large debris avalanches up to several km³ (sector or flank collapse, Blahůt et al., 2019).

Since the dramatic flank collapse of Mount St Helens in 1980 (Voight et al., 1983), massive debris avalanches (more than $10^7$ m³) have been widely studied (e.g. Siebert et al., 1987; Voight, 2000; Boudon et al., 2007). Siebert (1984) estimates a worldwide occurrence frequency of about 4 such events per century. Site specific occurrence frequencies are variable. For instance, the Soufrière de Guadeloupe volcano has produced at least 8 flank collapses in the past 9,150 years (Komorowski et al., 2005; Legendre, 2012). With one event per 1,100 yr on average, flank collapse events appear rather frequent on a
geological time scale. However, from a human perspective this represents only one event in 45 generations.

     In comparison, smaller mass wasting events (from $10^6$ to $10^7$ m³) are more frequent. Landslide occurrence rate $f$ is indeed related to landslide volume $V$ through a power law $f \propto V^{-\beta}$, with $\beta$ a positive parameter. For instance, Brunetti et al. (2009) estimate $\beta = 1.3$ for a global dataset of landslides ranging from $10^{-4}$ to $10^{13}$ m³ (including among others rockfalls, rock avalanches and debris avalanches). When considering landslides in volcanic context only, Brunetti et al. (2009) compute $\beta =$
1.1 (see their dataset R). Thus, although mass wasting events involving $10^6$ to $10^7$ m³ affect smaller areas in comparison to larger volcanic flank collapse (>$10^8$ m³), their higher occurrence rate may result in similar risk levels.

     In volcanic context, events involving $10^6$ to $10^7$ m³ span a wide variety of landslide type, including slow moving landslides (a few cm yr$^{-1}$ to a few m yr$^{-1}$) as in the Salazie cirque in La Réunion island, France (Belle et al., 2014; Rault et al., submitted), partial dome collapse during eruptive episodes (Harnett et al., 2019), and very fast (several tens of m s$^{-1}$) rock or
debris avalanches during or long after eruptive episodes (e.g. 2010 Mount Meager landslide in Canada, Guthrie et al., 2012; Moretti et al., 2015). In this study we focus on such rock and debris avalanches. They can be significant threats to populations and infrastructures due to their velocity and the difficulty to predict their occurrence. For instance, 2500 people were killed by the $1.6 \times 10^6$ m³ Casita (Nicaragua) debris avalanche in 1998 during hurricane Mitch (van Wyk de Vries et al., 2000; Scott et al., 2005).

In order to assess the risks associated to debris and rock avalanches, their propagation and the associated impact on populated areas can be estimated through numerical modelling (e.g. McDougall, 2017; Peruzzetto et al., 2019) or empirical laws (e.g. Mitchell et al., 2019; Peruzzetto et al., 2020). However, a major difficulty in such studies is the characterization of the initial unstable mass, that is, the initial conditions of propagation simulations. The initial unstable mass is characterized by its volume and/or its geometry. Both can be determined from the surface envelope separating the stable and unstable materials, and
corresponding to potential rupture surfaces. This envelope can be deduced from the extent of the landslide on the topography and relative simple geometric considerations, provided the subsurface is homogeneous (Jaboyedoff et al., 2019, 2020). The determination of the landslide extent is a problem by itself: it can be inferred, for instance, from field observations or records of surface displacements (e.g. with satellite monitoring, Schaefer et al., 2019; Hickey et al., 2020). However, the sub-surface is often not homogeneous in volcanic environment, because different volcanic materials were deposited during successive
eruptions with erosion and weathering phases in between. To construct the surface envelope, it is thus necessary to identify the different geological units composing the unstable slope. This requires field work and/or geophysical surveys (Rosas-Carbajal et al., 2016, 2017). The nature and geometry of geological units can then be processed in a expert way to identify preferential



rupture surfaces. Rupture surfaces can also be inferred from limit equilibrium analysis, provided geotechnical data is available (Apuani et al., 2005; Verrucci et al., 2019; Heap et al., 2021).

However, in many cases, researchers and/or practitioners can only rely on remote observations and/or topographic models to estimate the surface envelope of the unstable mass. This happens when field work is difficult or dangerous (e.g. in remote and steep areas), and when advanced remote sensing methods are not applicable (e.g. InSAR acquisition for displacement measurements does not yield conclusive results in densely vegetated areas). Besides, in volcanic context, simple geometric considerations linking the rupture surface to the extent of the unstable mass may not be applicable either (Jaboyedoff et al.,

2019, 2020): the volcanic formations below the surface are often not homogeneous and can display complex geometries. In these conditions where relatively few data are available but stakes require a quantified hazard assessment, how can we estimate the surface envelope of an unstable mass, from which the unstable volume can be quantified?

In this work, we show how the combined use of historical ortho-photographs and aerial photographs, Digital Elevation Models (DEMs) and 3D point clouds can help estimate the surface envelope of an unstable mass. We choose the Samperre

cliff in Martinique (Lesser Antilles, French West Indies) as a case study. Located on the western flank of the Montagne Pelée volcano, it has undergone several episodes of destabilizations since at least 1980 or even 1950 (Aubaud et al., 2013; Clouard et al., 2013). Although the resulting rock avalanches do not threaten directly populated areas, subsequent debris flows do propagate several kilometers downstream and impact populations, buildings and infrastructures. The quantification of potentially unstable volumes is thus important to assess the volume of the resulting loose debris reservoir that could feed

debris flows.

The geological context of the study site is given in Section 2. In Section 3 we present the topographic surveys and ortho-photographs used in this work, along with the methods used to (i) characterize the geometry of geological units and (ii) compute the volume of the unstable mass. Then, in Section 4, we describe the different geological layers forming the Samperre cliff and differentiate between stable and unstable layers. This allows to infer the paleo-morphology of the site and identify a

paleo-valley. We extrapolate the geometry of this paleo-valley to construct a stable basal surface above which materials can be remobilized. The associated volume is computed and discussed in Section 5, along with the other results of our work.

## 2 Study site

A detailed review of Martinique island geological history can be found in Westercamp et al. (1989); Maury et al. (1991); Boudon et al. (2005); Germa et al. (2011); Boudon and Balcone-Boissard (2021). The Martinique island is part of the Lesser

Antilles volcanic arc (Figure 1). The Samperre cliff is located in the northern part of the island, in the most recent volcanic formations and constituting since 550 kyrs the Montagne Pelée Volcanic Complex (Boudon and Balcone-Boissard, 2021, Figure 1a and).

The first stage of Montagne Pelée Volcanic Complex is a succession of andesitic breccias, lava domes and lava flows dated between 550 and 127 kyrs by Germa et al. (2011). The end of the first stage is marked by the Prêcheur flank collapse (Prêcheur

collapse structure in Figure 1, Le Friant et al., 2003; Boudon and Balcone-Boissard, 2021). This $25 \times 10^9$ m$^3$ collapse (Brunet





et al., 2016) entailed a lithostatic decompression of the lava chamber feeding this primitive Montagne Pelée edifice (Germa et al., 2011). It may have triggered the formation of the Piton Marcel lava dome near the boundary of the horse-shoe shaped collapse structure (Figure 1b).

The Prêcheur flank collapse marks the the beginning of the second phase of the Montagne Pelée Volcanic Complex. Lava dome-forming eruptions and associated concentrated pyroclastic density currents were the dominant activity. The edifice that grew in the Prêcheur collapse scar was destroyed by another flank collapse about 36 kyrs ago (Rivière Sèche collapse, Solaro et al., 2020). During the third stage of the Montagne Pelée Volcanic Complex, the first 10 kyrs years were characterized by abundant explosive activities with low-silica andesitic magma. Over the past 25 kyr, Boudon and Balcone-Boissard (2021) have recorded at least 55 magmatic eruptions, two third of which are dome forming eruptions, and the rest being plinian eruptions.

The volumes of flank collapses can be estimated from the on-land and submarine topography. However, before the new interpretation of Solaro et al. (2020), previous studies suggested the Prêcheur flank collapse 127 kyrs ago was constituted of two separate collapses, 127 kyrs and 32 kyrs ago (Le Friant et al., 2003; Germa et al., 2015; Brunet et al., 2016). Their volumes were estimated to $25 \times 10^9$ m$^3$ and $13 \times 10^9$ m$^3$ by Brunet et al. (2016). Smaller volumes were computed by Germa et al. (2015) using a geomorphological analysis: $14.7 \times 10^9$ m$^3$ and $8.8 \times 10^9$ m$^3$. Thus, the total volume for the Prêcheur collapse can be estimated between $27.5 \times 10^9$ m$^3$ and $38 \times 10^9$ m$^3$. However, these volumes were likely destabilized by successive smaller debris avalanches less than $5 \times 10^9$ m$^3$, as suggested by numerical simulations (Brunet et al., 2017). Such volumes are coherent with the volume estimated for the Rivière Sèche flank collapse, between $2 \times 10^9$ m$^3$ (Brunet et al., 2016) and $3.5 \times 10^9$ m$^3$ (Germa et al., 2015). Even so, the magnitude of such events is about 100 to 1,000 times larger than that of the rock avalanches we focus on (less than $10^7$ m$^3$).

Such rock avalanches occur in the Prêcheur river catchment that drains part of the western flank of Montagne Pelée. The Prêcheur river's major affluent is the Samperre river. The Samperre river skirts Piton Marcel to the south and has its source at the toe of Samperre cliff. This cliff is very recent, as it was formed between 1951 CE (when only smooth slopes are visible on ortho-photographs) and 1980 CE (first monitored collapse) by successive retrogressive failures (Figure 2). Five major destabilization sequences occurred in 1980, 1997-1998, 2009-2010 and 2018 (Aubaud et al., 2013; Nachbaur et al., 2019), but another unconfirmed collapse episode may have happened in the 1950s (Aubaud et al., 2013). Thus, the cliff rim has retreated by 250 m between 1988 and 2018 along a West-South-West / East-North-East direction (azimuth N062°E, white line in Figure 2a), resulting in a 300 m high steep ($> 60°$) to sub-vertical wall. Clouard et al. (2013) estimated that a volume of about $2.1 \times 10^6$ m$^3$ collapsed between March and May 2010. Between May 2010 and August 2018, the cliff lost about $4.9 \times 10^6$ m$^3$, with a major collapse sequence between January and August 2018 (Quefféléan, 2018; Nachbaur et al., 2019; Peruzzetto, 2021b; Peruzzetto et al., 2022). These destabilizations are often (if not systematically) associated with the remobilization of deposits and the generation of debris and hyper-concentrated flows. For instance in June 2010, a major debris flow inundated part the Prêcheur village at the river mouth and severely damaged the bridge crossing the river (Aubaud et al., 2013; Peruzzetto et al., 2022). A new and higher bridge has since been constructed, but major lahars could still destroy it. As a result, about 420 people would be isolated from the rest of the island (INSEE, 2015). For a local risk management perspective, it is thus important to



assess the volumes of future rock avalanches from the Samperre cliff in order to estimate the magnitude of the associated debris
flows and to quantify the risk for exposed population.

## 3    Material and methods

Our objective is to assess the surface envelope of potentially unstable volumes from the Samperre cliff. To that end, we need
to the geology of the cliff and infer the geometry of geological units. As field work is too dangerous and because we have no

geophysical data, we must only rely on remote observations and topographic surveys.

### 3.1    Data

Over the past decades, and in particular since the first documented collapse sequence in 1980, numerous oblique aerial pic-
tures taken during helicopter surveillance flights have documented the evolution of the cliff. We use them with georeferenced
ortho-photographs. The oldest available ortho-photographs date back to 1951 (IGN, 2021a, b). The next one were acquired in

1988 and then more regularly until 2018. The date and details of acquisitions are given in Table 1. The comparison between
successive photographs document the plan-view recession of the cliff and its geology. We analyze in details the 08/2018 ortho-
photograph to identify visually the different geological units composing the Samperre cliff from color or texture variations.

Several DEMs and 3D point clouds are also available, derived from LiDAR or photogrammetric acquisitions between March
2010 and August 2018 (see Table 1). They are used to characterize the geometry of identified geological units by means of the

CloudCompare software (CloudCompare, 2020). We characterize (i) deposition horizons identified by sharp color contrasts,
(ii) interfaces between units inferred from slope breaks and (iii) the surface of outcrops.

### 3.2    Geometric characterization of geological layers

We do not use the 08/2018 ortho-photograph to characterize deposition horizons because the Samperre cliff displays steep and
even over-hanging topographic features. Instead, we work directly on textured 3-D point clouds that allow a better charac-

terization of geometric features (e.g. Pavlis and Mason, 2017; Buckley et al., 2019). The horizons are thus identified on the
07/2010 photogrammetric 3D point cloud. This point cloud has the best resolution compared to other point clouds of the area
(on average, 32 pts/m$^2$ on the cliff). Horizons are picked in CloudCompare (CloudCompare, 2020) with the *Compass* plug-
in (Thiele et al., 2017). This plug-in automatically interpolates paths between manually picked points, with a least-cost-path
solver, provided a cost function. As deposition horizons are identified by color change, we choose a cost function that depends

on RGB (Red Green Blue) color gradient: the resulting path is chosen such that it follows strong RGB gradients. We pick
horizons that are first identified by visual inspection of the point cloud. *Compass* is also used to pick interfaces marked by a
slope break: in this case, the cost function depends on the point cloud local curvature.

To assess the dip and dip direction of the deposition horizons and interfaces, we sample points along the picked polylines
(10 pts/m). The resulting point cloud is then fitted by a plane. The quality of the fit is given by the Root Mean Square (RMS)

of the distances from the points to the best-fit plane. Following Fernández (2005), we also compute two indicators, $M$ and $K$





**Figure 1.** Map of the study site. (a) Geological map of Montage Pelée volcano, adapted from Westercamp et al. (1989) using the interpretation of Boudon and Balcone-Boissard (2021). The main phases of volcanic activity are: first stage / Primitive Montagne Pelée (550-127 kyrs), Second stage / Intermediate Montagne Pelée (126-36 kyrs) and Third stage / Recent Montagne Pelée (35-26 kyrs, and 25 kyrs-present). Older formations and alluvial deposits are not mapped. The insert is Martinique island. The red rectangle in insert gives the extent of (a). The white rectangle in (a) gives the extent of (b). (b) Close-up on the North-West flank of Montagne Pelée, with the Samperre Cliff, Piton Marcel, Morne Macouba, the Prêcheur and Samperre rivers, and main ravines. Ravines are are referred to with letters A, B, C and D in the text. The destabilization structures are reproduced from Boudon and Balcone-Boissard (2021)




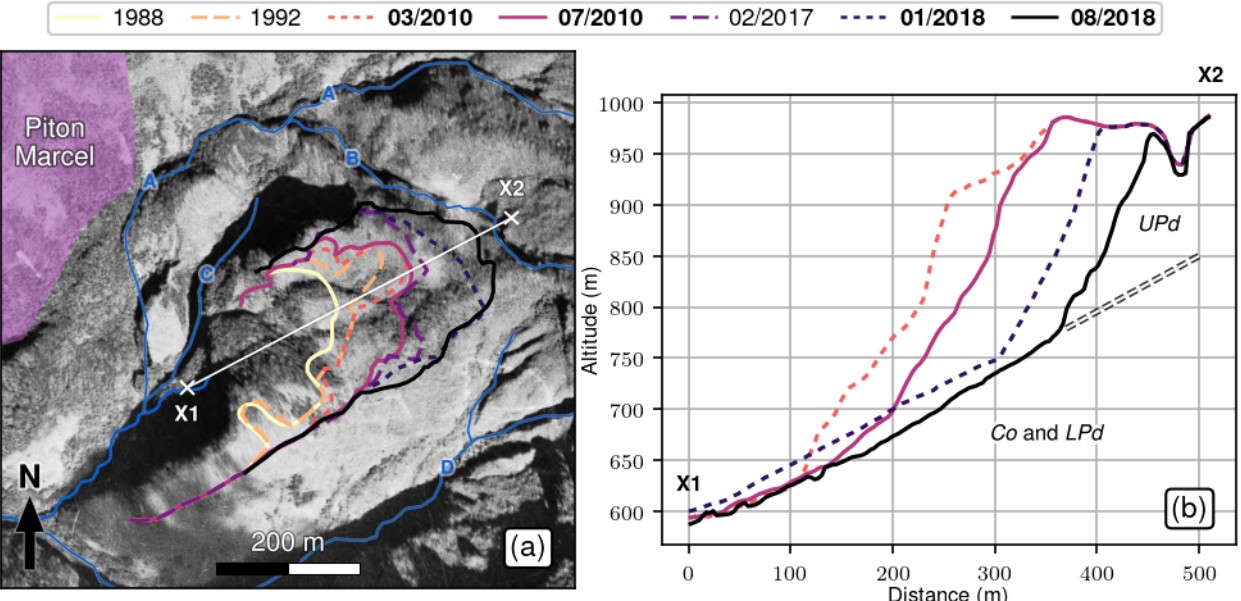

**Figure 2.** Evolution of the Samperre Cliff between 1951 and 2018. (a) 1951 ortho-photograph of the Samperre cliff. The successive cliff rims observed on ortho-photographs and DEMs between 1988 and 2018 are given by plain and dashed lines. The Samperre river (thick blue line), ravines (thin blue lines) and the Piton Marcel (purple patch) are given as in Figure 1. (b) Successive cross-sections of the cliff (03/2010, 07/2010, 01/2018, 08/2018, in bold in the legend). The inferred contact between the stable unit *UPd*, and unstable units *Co* and *LPd* is given by the white dashed line. The extent of the cross-section is reported in (a) by the segment [X1, X2].

**Table 1.** Ortho-photographs and topographic surveys.

| | Date | Resolution / Précision | Use | Source | Comment |
|---|---|---|---|---|---|
| **Ortho-photograph** | 1951, 1988, 1992 03/2010, 02/2017 | 0.5 m/pixel | - Cliff rim position | BD ORTHO®, BD ORTHO®Historique (IGN, 2021a, b) | n.a. |
| | 07/2010 | 0.2 m/pixel | - Cliff rim position | Ortho Hélimap/DEAL 972 | n.a. |
| | 08/2018 | 0.05 m/pixel | - Geological units | Ortho Hélimap/DEAL 972 | n.a. |
| **DEM derived from LiDAR** | 03/2010 | 1 m/pixel | - Cliff rim position - Unstable volume quantification | IGN Litto3D® | Acquisition over the whole Martinique island |
| | 08/2018 | | - Cliff rim position - Basal surface for unstable volume quantification | Hélimap company | Acquisition limited to the Samperre cliff, Samperre river and Prêcheur river |
| **Photogrammetric model** | 07/2010 | 32.3 pts/m$^2$ | - Cliff rim position - Basal surface for unstable volume quantification - Sedimentary interfaces | Pictures: Hélimap company 3D model: this study, with Agisoft Metashape | n.a. |
| | 01/2018 | 22.5 pts/m$^2$ | -Cliff rim position | Pictures: Carige company 3D model: this study, with Agisoft Metashape | n.a. |





(see Appendix A for computation details). Fernández (2005) suggest that $M > 4$ indicates a good fit between the plane and the point cloud, and that $K < 0.8$ indicates a correct estimation of the plane orientation. Sampling bias results in an estimation uncertainty on dip and dip direction. We quantify this uncertainty by computing the 100-times bootstrapped standard deviation of dip and dip direction (see Appendix A for details).

The same methodology is used to estimate the dip and dip orientation of outcrop surfaces extracted from point clouds.

### 3.3  Unstable volume quantification

The identification of the different geological units, the characterization of their geometry, and their evolution since 1950 evidences the progressive unearthing of interfaces limiting the basal and lateral surface of the successive destabilizations. Thus, we can differentiate between geological units affected by destabilisations and geological units that are stables. These
observations are used to reconstruct the paleo-morphology of the site and infer a basal stable surface above which materials are unstable.

This basal surface is composed of (i) outcrop surfaces of geological units considered stables (approximated by planes, as described in the previous section) and (ii) planes fitted manually in CloudCompare to topographic features that have remain stable since 1951. The potentially unstable volume is then defined as the rock mass contained between this basal surface, and
the topography (08/2018 DEM for the cliff, 03/2010 DEM elsewhere). Given the 1 m accuracy of the DEM, we consider only height difference superior to 1 m, in an area above the contact between the stable and unstable geological units.

The computed rock mass volume is dependant upon the different modelled planar units. To estimate volume variations caused by sampling bias, we randomly resample 100 times the outcrop surfaces of stable geological units and derive the associated best-fit planes and resulting basal surface. Then, we compute the bootstrapped standard deviation of the potential unstable
volume.

## 4  Results

In this section (i) we describe geological units composing the Samperre cliff, (ii) we identify which units are stable are which units are unstable, (iii) we reconstruct the paleo-morphology of the site, and (iv) we use these results to reconstruct the surface envelope of the unstable mass and compute the corresponding volume.

### 4.1  Geological description of the cliff

The different geological units composing the Samperre cliff (Figure 3) have been progressively exposed by successive destabilization episodes. In the following we describe the successive layers from top to bottom.

The top most layer of the cliff is composed by brown to light brown, probably weathered materials (unit *Pu*, yellow patch in Figure 3c). It was exposed in 2010 as a 20 to 30 meter thick layer (Figure 4a), as mentioned by Mathon and Barras (2010)
and Clouard et al. (2013). The 2018 destabilizations removed most of this geological unit. By March 2018, the corresponding layer at the cliff head was only about 10 meter thick (Figures 4b and 4c).



Below unit *Pu*, a massive, light gray to light brown, 50 m thick layer was exposed in 2018 to the East and South of the cliff (unit *La*, blue patch in Figure 3c). This layer displays clear vertical prismatic patterns. We associate these patterns to an andesitic lava flow (Figures 4b and 4c).

Most of the cliff below is formed by a 100 to 200 m succession of pyroclastic deposits (unit *UPd*, purple patch in Figure 3c). They include several meters wide blocks. The interface between successive deposits can be clearly identified by color changes, from light to dark gray (Figure 5a). On the 07/2010 photogrammetric model, we approximate these interfaces by planes (Figure 5b), with mean dip 16° and dip direction N254°E (see Table A1). This is, to within 15°, the opposite direction of cliff retreat since 1950 (N062°E, black dashed line in Figure 5b).

The bottom part of the cliff is composed, in its North-West side, by a characteristic ocher unit (unit *Co* in Figure 3c). Unit *Co* was initially unearthed in July 2010 (Figure 4a, surface *S1* in 3). Following the destabilizations of 2018, the initial outcrop extended North-Eastward (Figures 4c and 4d, surface *S2* in Figure 3). The outcrop surfaces *S1* and *S2* have, respectively, dip angle 47° and 42°, and dip direction N308°E and N252°E. Both surfaces do not correspond to any known tectonic fault. The transition between units *Co* and *UPd* is marked by a clear slope break (see Figure 3b and Figure 3c), and water seepages. An-

other ocher outcrop can be observed on the western side of ravine A, at the bottom of Piton Marcel (black patch in Figure 3c). Because unit *Co* has a distinctive ocher color, seems more consolidated and impermeable, we interpret it as old hydrother-malized volcanic materials (e.g. Salaün et al., 2011). As the northern extremity of the Prêcheur collapse scar is located only a few hundred of meters away from unit *Co*, we make the hypothesis that the surface of unit *Co* is associated to the Prêcheur flank collapse. It could either be the collapse scar itself, or the surface of a mega-block that slid and stopped in the vicinity of

the source area (Figure 6a). In this perspective, unit *Co* corresponds to eruptive materials emplaced during the first stage of Montagne Pelée Volcanic Complex.

     On the opposite side of the cliff, to the South-East, unit *UPd* lies on top of another succession of pyroclastic desposits, light gray, more indurated with no evidence of destabilizations or erosion since 2010. This unit, that we call the lower pyroclastic unit *LPd* (orange patch in Figure 3c), may have been already visible in 2010 in the South-East part of the cliff, but was more

obviously visible after the 2018 destabilizations (Figure 4c and 4d). The visible outcrop of *LPd* (surface S3 in Figure 3) is roughly planar, with dip angle 47° and dip direction N308°E, which does not correspond to any known tectonic fault. As for unit *Co*, the transition between *UPd* and *LPd* units is marked by a clear slope break (see Figure 3b and Figure 3c, black dashed line), accompanied by several water seepages. Another clear slope break can be observed on the other side of the cliff in the adjacent valley (ravine D). If we consider the areas at the junction between units *LPd* and *UPd* (i.e., black dashed lines

in Figure 3b and Figure 3c), both slope breaks are roughly coplanar, and the associated plane *S0* has dip angle 24° and dip direction N252°E. This is consistent with horizons measured in unit *UPd* (compare red and black points in Figure 5).

     Absolute dating is difficult without rock samples, as the Samperre cliff area is unsafe to wander through. Following our assumption that unit *Co* is associated to first stage of Montagne Pelée Volcanic Complex phase, it may be at least 127 kyrs old. Pyroclastic deposits within the collapse scar (units *LPd*) are more recent and could have been emplaced in between the

Prêcheur flank collapse (127 kyrs) and the Rivière Sèche collaps (36 kyrs), as suggested by the geological map of Westercamp et al. (1989) and the interpretation of Boudon and Balcone-Boissard (2021). The upper pyrocalstic despoits units *UPd* would



**Figure 3.** Geological interpretation of the Samperre cliff. (a) 08/2018 orthophotograph (Hélimap/DEAL 972), with contours of identified geological units. (b) Slopes of the 08/2018 DEM, completed with 07/2010 DEM , with contours of identified geological units. (c) 08/2018 DEM, completed with 07/2010 DEM. Colored patches correspond to geological units: *Co*, *LPd*, *UPd*, *La*, *Pu* (see main text for details). Hatched areas identify some of the surfaces used to construct a potential unstable volume (see Section 4.4). The contact between *LPd* and *UPd* units, inferred from slope breaks, is given in (b) and (c) by the black dashed lines. Ravines and rivers are also displayed, as in Figure 1.

then correspond to the eruptive phase characterized by dome-forming and plinian eruptions (starting 25 kyrs ago, Boudon and Balcone-Boissard, 2021). The origin of the lava flow unit *La* is uncertain, but could be associated with the formation of the neighboring Morne Macouba 12 kyrs ago (Germa et al., 2011). Following Mathon and Barras (2010) and Clouard et al. (2013), we interpret unit *Pu* as pumices emplaced by the plinian eruption P1 that occurred between 1289 and 1398 CE (Boudon et al., 2005).





**Figure 4.** Aerial photographs of the Samperre cliff on (a) Jun. 25, 2010, (b) Jan. 2, 2018, (c) Mar. 8, 2018 and (d) Jul. 2, 2018. East direction is given by the arrow, the scale is given for the cliff rim. Outcrops of visible geological units are highlighted.



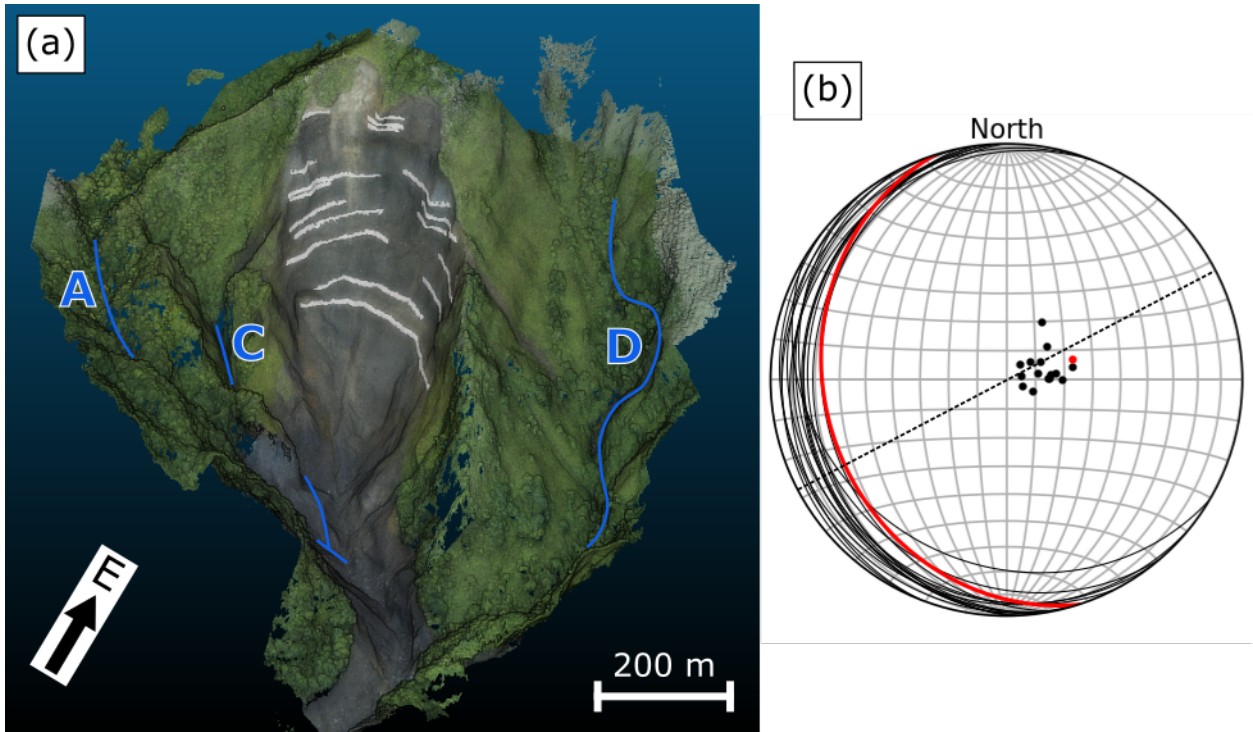

**Figure 5.** Deposition horizons orientation. (a) July 2010 photogrammetric 3D model, with deposition horizons (white lines). Ravines are highlighted by blue lines with their name, as in Figure 1. (b) Wolff stereonet projection of deposition horizons (black lines), with corresponding poles (black dots). The contact *S0* between LPd and UPd is also given in red. The black dashed line gives the cliff retreat direction (azimuth N062°E, white line in Figure 2)

## 4.2 Identification of stable and unstable geological units

From successive ortho-photographs and topographic surveys (Figure 2), as well as aerial photographs (e.g. Figure 4a and 4b), it is clear that the upper geological units *Pu*, *La* and *UPd* are involved in the main destabilizations episodes. Thus, we conclude

they are unstable. Following NGU (2012), the presence of water seapge at the base of unit *UPd* also favors its instability.

Units *Co* and *LPd* were previously recovered by unit *UPd*. Successive destabilizations have increased the surface of the outcrops. These outcrops do not display any clear collapse scar. They have remained relatively intact since they were exposed (e.g. compare outcrop of unit *Co* in Figures 4a and 4c). This suggests units *Co* and *LPd* are more resistant than unit *UPd*. This stability is further confirmed by water resurgences at the base of unit *UPd*, which indicates that the stability of units *Co* and

*LPd* is less affected by increased pore pressures.

Thus, we make the hypothesis that units *Co* and *LPd* are stable. From this assumption we deduce that the outcrop surfaces *S1* and *S2* (respectively *S3*) correspond to the formerly hidden interface between unit *Co* (respectively *LPd*) and unit *Upd*.



### 4.3 Paleo-morphology

From the geological and geometric observations described above we propose a scenario of the evolution of the catchment
geomorphology, with successive construction and dismantling phases.

Following the interpretation of Solaro et al. (2020), the Prêcheur flank collapse that occurred 127 kyrs ago is the oldest
event that can be related to the current morphology of the Samperre cliff. This major dismantling event was followed by a
construction phase. The accommodation space left by the collapse was progressively filled with new pyroclastic deposits (unit
*LPd*, Figure 6a). Its surface may be given by the current interface between units *LPd* and *UPd* (surface *S0*).

A new dismantling phase then started. It may have been initiated by the Rivière Sèche flank collapse 36 kyrs ago: this
collapse did not affect the Samperre cliff area but the resulting scar concentrated newly emitted volcanic materials. Thus, unit
*LPd* was progressively eroded. We suggest erosion was mainly caused by preferential flow path along the Prêcheur collapse
structure, and thus along unit *Co*. It led to the formation of a valley in a West-South-West/East-North-East direction (N062°E,
Figure 6b). The North-West flank of the paleo-valley is given by surfaces *S1* and *S2* of unit *Co*, and the South-East flank by
surface *S3* of unit *LPd*.

The emplacement of pyroclastic deposits from unit *UPd* correspond to a new construction phase, when pyroclastic materials
started filling the paleo-valley (Figure 6c). Following Boudon and Balcone-Boissard (2021), this phase possibly started 25 kyrs
ago. Because deposition was geometrically constrained by the valley, successive deposition horizons from unit *UPd* have dip
direction similar to that of the paleo-valley.

Then, a new, still ongoing, erosion phase started. As a result of permeability contrast between basal units *Co* and *LPd*, and
upper unit *UPd*, underground water flowed preferentially at the interface between these units, that is, along the paleo-valley
surface (as evidenced, today, by water seepages). It resulted in the progressive drainage of the paleo-valley by weakening
the base of unit *UPd*, and thus favoring destabilizations. This process eventually led to the formation of the Samperre cliff
(Figure 6d). It progressively retreated in the same direction as the paleo-valley. However, the erosion phase certainly started
long before the known historical destabilizations, possibly as soon as the North-West flank of Montagne Pelée was no longer
fed by eruptive materials.

As of 2022, it is still an ongoing process. Thus, we suggest that the volumes that could be involved in future rock avalanches
are constrained by the volume of materials filling the paleo-valley. To quantify this volume, we reconstruct the surface of the
paleo-valley that we believe is stable.

### 4.4 Assessment of unstable volume

The geometry of the paleo-valley is reconstructed by extrapolating outcropping surfaces of stable units and surfaces fitted to
stable topographic features (see Section 3.3 for methodological details). These surfaces are displayed in Figure 7a, and their
characteristics are given in Table 2.

The North-West (respectively South-East) side of the paelo-valley,is constrained by the surfaces *S1* and *S2* (respectively
*S3*) of unit *Co* (respectively unit *LPd*). We also fit manually a planar surface *S4* to the North-West wall of ravine B, in the





**Figure 6.** Conceptual view of the evolution of the paleo-morphology of the Samperre cliff, along a North-West (left) / South-East (right) cross-section (*MM'* in Figure 1b). (a) Emplacement of unit *LPd* within the Prêcheur collapse scar. (b) Erosion of unit *LPd*, resulting in a paleo-valley at the location of the actual Samperre cliff. (c) Emplacement of unit *UPd*. (d) Erosion of unit *UPd*, resulting in the formation of the Samperre cliff and adjacent ravines. Black plain line correspond to the topography given by the 08/2018 DEM. White dashed line are interpretations of old topographies and contact between geological units. See Figure 1c for the position of ravines.





**Figure 7.** Construction of potential basal surface for unstable volume quantification. (a) Surfaces used to define the basal surface. S1 to S4 are point clouds extracted from DEMs, S5 and S6 are planes manually fitted to topographic trends (see Table 2). (b) Basal surface (red planes), constructed by extending and combining planes fitted to S1, S2, S3 and S4, and planes S5 and S6. The Piton Marcel is highlighted by the purple patch. The black dashed line is the Prêcheur destabilization structure.

continuation of which we identified another outcrop of unit *Co* (see Figure 3c). Similarly, we fit a planar surface *S5* following the southern ridge of Samperre cliff, that has remained stable since at least 1988. Finally, we assume that the bottom of the paleo valley matches the current bed of the Samperre river (surface *S6*).

The resulting basal surface envelope is represented in Figure 7b. The height difference between the basal surface and the 275 2018 topography is displayed Figure 8. The corresponding volume is about $8.3 \times 10^6$ m$^6$. The uncertainty associated with least square plane fitting (but not to manual plane fitting) amounts to less than 1% of the estimated volume, and is thus negligible (see Table 3).



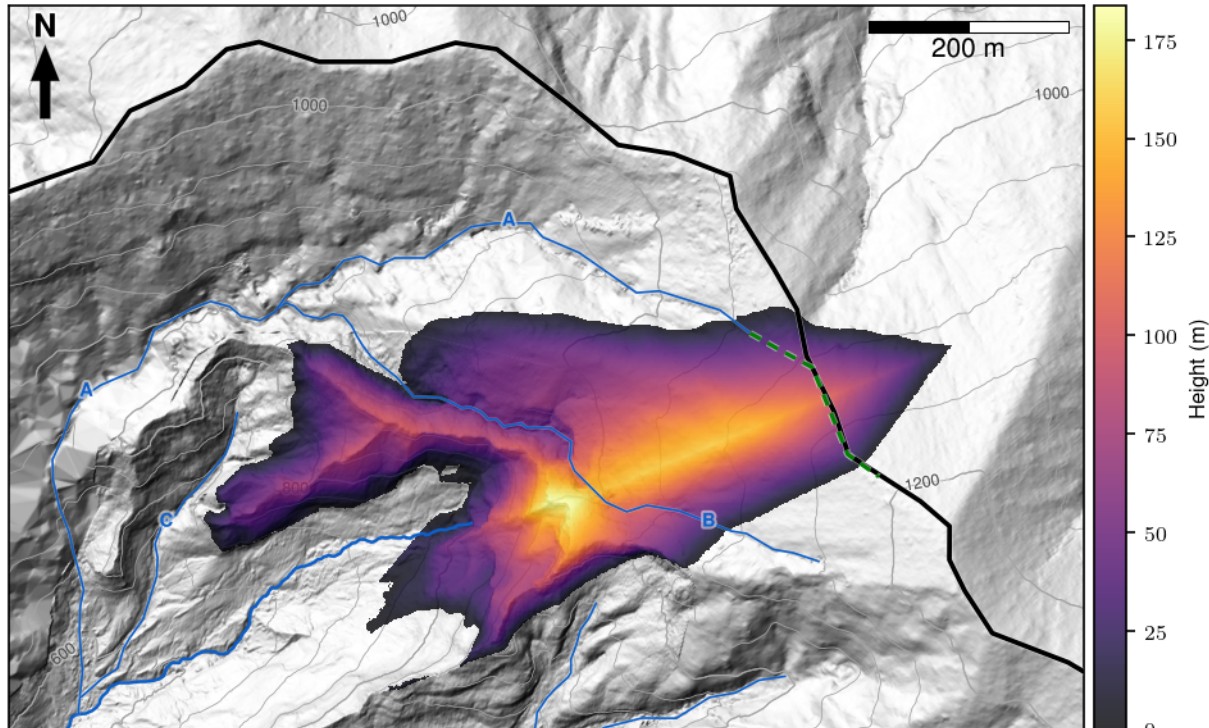

**Figure 8.** Potentially unstable volume. The height between the reconstructed basal surface (see Figure 7b) and the combined 07/2010-08/2018 DEM (displayed here) is given by the color scale. We only display heights above the contact between D and UPd units, and LPd and Upd units, and disregard heights below 1 m. The Samperre river (thick blue line), ravines (thin blue lines) and the Prêcheur destabilization structure (black line) are displayed as in Figure 1. The unstable volume is computed either directly, or by limiting the extent of the collapse (i) to ravine A continued along a topographic edge (green dashed line) or (ii) to ravine B (see Table 3).

**Table 2.** Surfaces used for quantifying a potential unstable volume, with characteristics of the associated best-fit planes. $\sigma$ is the standard deviation of estimated dip and dip directions. See Appendix A for characteristic lengths and widths computations, and for the derivation of $M$ and $K$. Following Fernández (2005), a good fit and reliability correspond to $M \geq 4$ and $K \leq 0.8$

|  | Description | dip (°) | | azimuth (°) | | RMS (m) | Characteristic length (m) | Characteristic width (m) | $M$ | $K$ |
|---|---|---|---|---|---|---|---|---|---|---|
|  |  | mean | $\sigma$ | mean | $\sigma$ |  |  |  |  |  |
| S1 | *Co* outcrop visible in 2010 | 42.17 | 0.03 | 168.75 | 0.06 | 1.79 | 34.88 | 6.96 | 5.94 | 1.19 |
| S2 | *Co* outcrop visible in 2018 | 43.92 | 0.02 | 205.70 | 0.04 | 1.69 | 44.23 | 16.57 | 6.53 | 0.43 |
| S3 | *LPd* outcrop | 47.32 | 0.02 | 308.00 | 0.01 | 7.23 | 161.76 | 29.90 | 6.21 | 1.19 |
| S4[1] | Northern extent | 57.32 | n.a. | 136.88 | n.a. | n.a. | n.a. | n.a. | n.a. | n.a. |
| S5[1] | Southern rim | 65.00 | n.a. | 318.00 | n.a. | n.a. | n.a. | n.a. | n.a. | n.a. |
| S6 | River bed | 11.95 | 0.00 | 243.26 | 0.05 | 1.94 | 124.08 | 30.65 | 8.31 | 0.51 |

[1] Planes fitted manually to fit topographic trends





**Table 3.** Potentially unstable volumes in m$^3$. $\sigma$ is the standard deviation of volume estimation, when considering the uncertainty on basal surface reconstruction (see Appendix A)

|  | Total volume (m$^3$) | South of ravine A (m$^3$) | South of ravine B (m$^3$) |
|---|---|---|---|
| Mean | $8.3 \times 10^6$ | $7.9 \times 10^6$ | $3.5 \times 10^6$ |
| $\sigma$ | $5.9 \times 10^3$ | $4.8 \times 10^3$ | $1.3 \times 10^3$ |

## 5 Discussion

### 5.1 Stability of unit *Co* and *Lpd*

We inferred from visual observations over more than a decade that units *Co* and *LPd* are more stable than unit *UPd*. In order to further investigate this hypothesis, we would need to characterize the geotechnical properties of these units. Geotechnical properties can be measured with in-situ tests (e.g. penetration tests) or with laboratory analysis of rock samples (e.g. tri-axial tests). However, both methods require field work that would be very dangerous near the Samperre cliff for safety reasons. In this context, it is more reasonable to rely on remote observations. For instance, the stability of units *Co* and *LPd* in coming
years could be assessed by monitoring their evolution, for instance with yearly visual observations or topographic surveys.

The stability of units *Co* may be questioned. Indeed its ochre colour suggests it has been extremely hydrothermal altered (e.g. Salaün et al., 2011), and many studies (e.g. Heap et al., 2021) have linked hydrothermal alteration to reduced rock stability. However, alteration may have affected unit *Co* only at its surface. It is indeed a preferential flow path, as evidenced by water seepage at the interface between units *Co* and *UPd*. Besides, even if the unit is altered at greater depth, these water seepages
suggest hydrothermal fluids no longer circulate within unit *Co*. Hydrothermal activity certainly stopped 127 kyrs ago, when the first flank collapse deeply impacted the primitive volcanic edifice of Montagne Pelée. Thus, we argue that the alteration of unit *Co* is old and has been compensated by (i) hydrothermal sealing, (ii) diagenetical cementation and (iii) lithostatic compaction (e.g. del Potro and Hürlimann, 2008). These three processes reduce connected porosity. For volcanic rocks, lower porosity is associated to increased cohesion and friction coefficient (Villeneuve and Heap, 2021). Thus, stability is increased, and the
alteration of unit *Co* is compensated for. In turn, the surface of unit *Co* becomes a preferential sliding surface as observed for instance for old collapse scars on the Soufrière de Guadeloupe volcano (Salaün et al., 2011).

There is no clear sign of hydrothermal circulation within unit *LPd*, which limits possibility of both alteration and hydrothermal sealing. However, the proximity of the Montagne Pelée eruptive center suggests pyroclastic deposits of unit *LPd* were emplaced at relative high temperature, which favours welding (Quane and Russell, 2005). Then, as for unit *Co*, unit *LPd*
was certainly further strengthened by diagenetical cementing and lithostatic compaction. Furthermore, Boudon and Balcone-Boissard (2021) describe most of the pyroclastic deposits produced during this second stage as indured materials forming a succession of ridges particularly visible on the more eroded western flank of Montange Pelée edifice.





## 5.2 Cliff destabilizations related to the drainage of a paleo-valley

The correlation between paleo-morphology and volcanic flank collapse has been investigated in several studies. For instance,
Branca and Ferrara (2013) relates past destabilizations of Etna volcano to the morphology of sedimentary units it grew on.
Similarly, over the past 10 kyr, numerous flank collapses from the Soufrière de Guadeloupe volcano have occurred repeatedly
in the collapse scar left by Morne Amic event (Komorowski et al., 2005; Rosas-Carbajal et al., 2016).

However, we could not find any study correlating directly destabilization of volcanic escarpments to the drainage of paleo-
valleys, in tropical context or anywhere else. For instance, Branca (2003) analyze the successive filling and drainage phases of
a fluvial valley affected by basaltic lava flows on Etna's flank, but do not associate it to destabilization episodes.

The Réunion island (Indian ocean) might be a good candidate to find analogues of the Samperre cliff paleo-morphology.
Indeed, Réunion island includes both an extinct and eroded volcanic center, the Piton des Neige, and an active volcano, the Piton
de la Fournaise. An example of paleo-valley infilled by volcanic materials is the upper reach of the Rivière des Remparts (see
figure 25.4 in Villeneuve et al., 2014). However, Piton des Neige and Piton de la Fournaise volcano have mainly erupted lava
flows (Salvany et al., 2012; Chaput, 2013). Besides, Rault et al. (submitted) suggest that most mass wasting events occurring in
the Cirque of Salazie depression (associated to the dismantling of Piton des neiges volcano) are associated to the remobilization
of old debris avalanches and epiclastic deposits (see also Chaput, 2013). As a result, destabilization mechanisms may differ
from the Samperre cliff that is mainly composed of pyroclastic deposits.

Without any documented similar case studies, further data are needed to confirm or invalidate the hypothese that the desta-
bilizations of the Samperre cliff are linked to the drainage of a paleo-valleys made in this study. In this perspective, it will be
important to analyze of future destabilization episodes, should they occur. In the mean time, the geometry of the paleo-valley
could be constrained by geophysical surveys. Indeed, given observed water seepages, the contact between units *Co* and *UPd*,
and *LPd* and *UPd*, is certainly associated to a permeability contrast, and thus a resistivity contrast (e.g. Romano et al., 2018;
Huntley et al., 2019). Indeed, older volcanic formations often tend to have lower permeabilities as a result of filling of cavities
and fractures (Singhal and Gupta, 2010), hydrothermal sealing (Polak et al., 2003) and compaction (Farquharson et al., 2017).

Given the difficulty to carry out terrestrial surveys in the Samperre cliff area, airborne electromagnetic surveys (AEM) are
more feasible. They have already proven to provide valuable information to constrain landslides geometry (Nakazato and
Konishi, 2005). Most of Martinique island was covered in 2013 by AEM surveys (Deparis et al., 2014; Coppo et al., 2015), and
the resulting data was used to investigate hydrogeological systems (Vittecoq et al., 2015, 2019) and active landslides (Thiery
et al., 2017, 2021). This highlights the great potential of this method, but unfortunately the Samperre cliff could not be flown
over in 2013. The current development of AEM drone-based survey could help acquiring AEM data on this zone. We expect the
contrast between *Co* and *UPd* to be particularly clear, as these units clearly have different lithologies or degree of weathering
(as evidenced by color variations). However, unit *LPd* and *UPd* are both pyroclastic deposits, hence the contrast may be less
marked but still identifiable because the difference in age of deposition and consolidation.



### 5.3 Volume estimation and implication for hazard assessment

The main surfaces controlling the extent of the reconstructed paleo-valley behind the cliff are *S2* and *S5*. The river bed (surface *S6*) has a very limited influence, because it lies below other surfaces, and in particular below *S1*, *S2* and *S3*. This is why it does not appear in Figure 7b). Similarly, surface *S3* does not influence the final computed volume, provided we consider only unstable volume in the Samperre cliff.

Our estimation of the uncertainty on the unstable volume is only related to the uncertainty on best-fit plane derivation. A more robust procedure should also take into account (i) the uncertainty on manual plane fitting for surfaces *S4* and *S5* and (ii) possible variations in structural orientations below the 08/2018 topography. In particular, we may expect the surface of unit *Co* to curve itself to the East, following the same trend as the Prêcheur destabilization structure, instead of keeping the orientation of surface *S2* (Figure 7b). This would be coherent with observations, as the dip direction of surfaces associated to unit *Co* gradually increases from the West to the East: N137°E for *S4*, N169°E for *S1* and N206°E for *S2* (see Figure 7 and Table 2).

Despite the lack of quantitative information, this possible bending of the interface between units *Co* and *UPd* can be taken into account, in a first approximation, if we simply delimit vertical surfaces constrained by topographic features. In a first case scenario, we assume that the paleo-valley did not extend further than the destabilization structure of the Prêcheur flank collapse. We thus use a North-East limit defined by ravine A (that follows the Prêcheur structure), continued along the ridge of the Prêcheur structure (green dashed line in Figure 8). The corresponding volume, $7.9 \times 10^6$ m$^3$, is of the same order than our first estimation ($8.3 \times 10^6$ m$^3$) to within 5% difference. In a second more conservative scenario, we use ravine B (that runs behind Falaise Samperre) as North-East limit. The associated volume is $3.5 \times 10^6$ m$^3$.

The upper volume value $8 \times 10^6$ m$^3$ gives an estimation of the total rock mass volume that could be involved in future destabilization episodes, until the catchment reaches a long-term stability state. Given the collapsed volume between 2010 and 2018 (about $5 \times 10^6$ m$^3$), this equilibrium state may well be reached in the coming decades. However, this does not entail that hazard associated to lahars will be then reduced in the Prêcheur river. Indeed, solid materials remobilized by lahars could also come from multiple other sources, including lateral erosion from the banks of the Samperre river and fresh euprive materials in the event of a new phreatic or magmatic eruption. Besides, although units *Co* and *LPd* are more stable than unit *UPd*, they may still be subjected to retrogressive erosion or gravitational instablities. Finally, gravitational instabilities could also occur in adjacent gullies, in particular in ravine D where superficial instabilities were witnessed in 2019 (figure 2 in Peruzzetto, 2021a).

In comparison, the smaller volume value $3.5 \times 10^6$ m$^3$ is consistent with volumes destabilized in recent times (e.g., $2.1 \times 10^6$ m$^3$ in 2010 and $4.9 \times 10^6$ m$^3$ between 2010 and 2018) and could correspond to the volume of the next destabilization episode. It could collapse in a single event or, more likely, in a succession of smaller rock avalanches. In both cases, a significant scree reservoir will be formed at the toe of the Samperre cliff. The processes controlling its remobilization depend on factors that are hard to constrain in real-time monitoring, including precipitation intensities and pore-pressure within the reservoir. The resulting lahars may occur as a few major high-discharge events with the potential to flood the Prêcheur village downstream, as in 2010; or as multiple smaller but still dangerous events during a longer time period, which is highly disruptive for inhabitants, as in 2018. Our study suggests either of these crisis situations could still occur in the coming years, as a result destabilizations





from the Samperre cliff. In the worst case scenario, we can consider a $3.5 \times 10^6$ m$^3$ reservoir created at the bottom of the
370 cliff (neglecting bulking), and remobilized in a single lahar with solid fraction 75% (following classical solid fraction values
as in Vallance and Iverson, 2015; Thouret et al., 2020). A rough estimation of the volume of the lahar is then $4.7 \times 10^6$ m$^3$.
The associated damages would be very important. Indeed, using numerical simulations, Peruzzetto et al. (2022) showed that a
$2.0 \times 10^6$ m$^3$ lahar has already the potential to flood a large part of the Prêcheur village.

However, in the current state of knowledge, it is impossible to predict the date of the next collapse and of the subsequent
lahars: the physical processes controlling and triggering destabilizations on the Samperre cliff are difficult to constrain. Never-
theless, they are most likely associated with water circulations.

### 5.4  Destabilization mechanisms

As already suggested by Clouard et al. (2013) and Nachbaur et al. (2019) ground water circulations at the interface between
units *Co* and *UPd* may be the main triggering factor for destabilizations. In the medium term (i.e. years), it could weaken the
380 base of unit *UPd* by washing out fine particles. This suffusion process is widely documented in the litterature (e.g. Moffat
et al., 2011; Wan and Fell, 2008; Prasomsri and Takahashi, 2021). Precipitations (and associated increased pore pressure)
could be a triggering factor in the short-term, but only after a time lag allowing ground water to reach the cliff. Indeed, the
2018 destabilizations episodes started only 2 days after the major rainfall of Dec., 31 2017 (almost 150 mm in one day in the
Prêcheur village Quefféléan, 2018)). We do not have access to precipitations records for previous events.

Interestingly, the chronology of recent destabilization episodes suggest that weakening by ground water circulations is par-
ticularly important at the interface between units *Co* and *UPd*. Indeed, during the 2009-2010 destabilizations episode (Clouard
et al., 2013), the first major collapses occurred on Aug. 19, 2009 on the North-West flank of the cliff (that is, in unit *UPd* above
unit *Co*). The western and southern side then collapsed less than a year after, on May, 11. 2010. Similarly, ortho-photographs
reveal that between 2010 and 2017, the cliff rim retreated to the north. The major 2018 destabilization episode concerned
mainly the western side of the cliff (Figure 2a). Thus, we suggest destabilizations espiodes start on the North-West side of the
cliff, at the interface between units *Co* and *UPd*, and progress to the East and South-East by successive retrogressive failures.

### 5.5  Contribution of rock avalanches to volcano dismantling

The volume detached from the Samperre cliff between March 2010 and July 2010 is about $2.1 \times 10^6$ m$^3$ (Clouard et al.,
2013), and $4.9 \times 10^6$ m$^3$ between July 2010 and August 2018 (Quefféléan, 2018; Peruzzetto, 2021b; Peruzzetto et al., 2022).
This yields an average volume of $0.9 \times 10^6$ m$^3$ yr$^{-1}$. This value certainly over-estimates the sediment production rate due to
rock avalanches, because the 2018 collapse sequence was particularly large compared to previous episodes in 1998 or 2010.
Quefféléan (2018) gives a rough estimation of the collapsed volume between 1951 and 2013 of $10 - 12 \times 10^6$ m$^3$, for an average
sediment production of about $0.16 - 0.19 \times 10^6$ m$^3$ yr$^{-1}$. Considering that $15 \times 10^6$ m$^3$ collapsed between 1951 and 2018, we
get a similar result: $0.22 \times 10^6$ m$^3$ yr$^{-1}$. It is twice as much as the estimation $0.1 \times 10^6$ m$^3$ yr$^{-1}$ of Clouard et al. (2013). If
we assume a density of 3000 kg m$^3$, we get a flux between 0.3 and 0.6 Mt yr$^{-1}$: it represents between 0.2% and 0.5% of the
sediment flux from cliff retreat in Europe (over a distance of 127,000 km, see Regard et al., 2022).



In comparison, the total volume volumes involved in the Prêcheur and Rivière Sèche flank collapses is estimated between $27 \times 10^9$ m$^3$ (Germa et al., 2015) and $40 \times 10^9$ m$^3$ (Brunet et al., 2016). The associated long-time annual averages are $0.21 \times 10^6$ m$^3$ yr$^{-1}$ and $0.31 \times 10^6$ m$^3$ yr$^{-1}$. Taking into account the on-land deposits of flank collapse, Germa et al. (2015) estimate
a slightly lower rate of volume evacuated at sea by flank collapses: $0.15 \times 10^6$ m$^3$ yr$^{-1}$. All these rates are of the same order of magnitude as sediment production by rock avalanches from the Samperre cliff.

Of course, the Samperre cliff has been particularly active since (at least) 1950 and our results suggest it could reach an equilibrium state in the coming decades. Thus, the extent to which the associated sediment production rate can be extended to longer time periods and to the whole Pelée edifice can be discussed. In particular, its specific location near the head-scarp of
the Prêcheur collapse structure could enhance instabilities in comparison to other escarpments. As suggested by Germa et al. (2015), it is important to differentiate erosion rates inside and outside collapse calderas. However, the pyroclastic materials composing most the cliff are, presumably, not specific to this site. Other outcrops are thus likely to be also eroded in the long-run by smaller landslides or runoff. This is particularly true in tropical islands where intense rainfalls favour superficial instabilities and erosion (e.g., more than 200 landslides cataloged in Martinique island between 2000 and 2020 Thiery et al.,
2021).

Thus, following Clouard et al. (2013), Germa et al. (2015) and Quartau et al. (2015), we suggest that the contribution of large edifice collapse to volcanic island dismantling is significant, but not necessarily dominant in comparison to other erosive processes. Of course, this may depend on the geological context and this question is still discussed in the litterature. For instance, Salvany et al. (2012) argue that the the formation of volcanic cirques in the Réunion island (Indian Ocean) was
mainly the result of regressive erosion. On the contrary, Chaput (2013) suggests that the excavation of the cirques was first initiated by major gravitational instabilities, whose resulting breccias were then eroded. More recently, Rault et al. (submitted) estimate that landslides triggered by rainstorms have contributed to about 20% of the erosion of the Piton des Neiges over the last 72 kyrs.

In our view, the long-term role of intermediate landslides (i.e. between $10^6$ and $10^7$ m$^3$) should be further investigated.
Indeed, landslides inventory often focus on smaller landslides or larger landslides, as can be clearly seen in figure 2 of Brunetti et al. (2009) giving the volume distribution of 19 landslide databases. Small landslides (below $10^6$ m$^3$) are indeed more frequent and can be thus cataloged at the local or regional scale, from historic reports and field investigations. On the contrary, larger landslides inventories (above $10^8$ m$^3$) are constructed from the geomorphological analysis of topographic surveys. Given the relative scarcity of such large landslides, large areas are investigated. At this scale, the footprint of intermediate
landslides is less clear, such that their comprehensive and systematic identification is not easy. Future research on that matter and the construction of comprehensive ladslide databases in volcanic islands (e.g. Rault et al., submitted), would improve our understandings of erosion mechanisms in volcanic islands.





# 6   Conclusions

The quantification of unstable volumes in rocky cliffs in volcanic context is difficult because of the complex geometry of
successive geological layers, especially when no geophysical, geotechnical or displacement data are available. In this study,
we used historic aerial oblique photographs, ortho-rectified photographs, Digital Elevation Models and 3D point clouds to:

1. Identify geological units and characterize the geometry of deposition horizons, interfaces and outcrop surfaces,

2. Identify stable and unstable geological units,

3. Reconstruct the paleo-morphology of the catchment, from which we deduced the 3D surface of a basal stable layer

4. Quantify the potential unstable volume for future destabilizations

We used the Samperre cliff (Martinique, Lesser Antilles), as a case study. Over the past decades it has produced several
rock avalanches from 2 to $5 \times 10^6$ m$^3$, whose deposits can be remobilized as debris flows threatening the Prêcheur village,
at the mouth of the Prêcheur river. From our observations and geometric characterization of geological units, we show that
destabilizations of the Samperre cliff could be related to the drainage of a paleo-valley. We estimate the volume of materials
still filling this paleo-valley to $7.9 \times 10^6$ m$^3$, and the volume involved in the next destabilization episodes to $3.5 \times 10^6$ m$^3$.
These results are important for operational risk management.

Our interpretation that destabilizations are linked to the drainage of a paleo-valley could be verified or contradicted by
airborne electro-magnetism surveys. They would identify, if it exists, the interface between the paleo-valley and materials
filling it. A regular monitoring of the cliff morphology during future destabilization episodes will also help better constrain
long-term evolution scenarios. More generally, our study contributes to quantify erosion rates in volcanic islands.

*Data availability.* The orthophotographs, DEMs, and photographs used to construct the photogrammetric models are not the property of the
BRGM. Thus, they can't be made freely available.

## Appendix A:  Geometric analysis of points cloud

Provided a set of $N$ points with coordinates $((X_1, Y_1, Z_1), ..., (X_N, Y_N, Z_N))$, we fit a plane of equation $aX + bY + cZ + d = 0$
to the corresponding cloud points through an ordinary least square regression, minimizing the distance between the points and
the plane. The associated minimal Root Mean Square ($RMS$) is:

$$RMS = \left( \frac{1}{N} \sum_{i=1}^{i=N} (aX_i + bY_i + cZ_i + d)^2 \right)^{\frac{1}{2}}.$$

To characterize the robustness of the fit, we introduce the $(N \times 3)$ matrix $M$:





$$M = \begin{pmatrix} X_1 - E(X) & Y_1 - E(y) & Z_1 - E(Z) \\ \vdots & \vdots & \vdots \\ X_N - E(X) & Y_N - E(Y) & Z_N - E(Z) \end{pmatrix},$$

where $E(X)$, $E(Y)$ and $E(Z)$ are respectively the mean $X$, $Y$ and $Z$ coordinates of the point cloud. The $(3 \times 3)$ covariance matrix $C$ is then given by:

$$C = \frac{1}{N} M^T M$$

    The three positive eigen-values $\lambda_1$, $\lambda_2$ and $\lambda_3$, and associated eigen-vectors $e_1$, $e_2$, and $e_3$ help describe the geometry of the point cloud. Assuming $\lambda_1 \geq \lambda_2 \geq \lambda_3$, $e_1$ points in the direction of the cloud main axis, $e_2$ in the second main axis direction

and $e_3$ in the third axis direction, all three axes being orthogonal to one another. The distribution of the points along these three axes is quantified by the eigen-values. For instance, if the points are aligned along a line, $\lambda_2 = \lambda_3 = 0$. If the points are homogeneously distributed on a disk, $\lambda_1 = \lambda_2$ and $\lambda_3 = 0$. In this perspective, $\sqrt{\lambda_1}$ can be seen as a characteristic length of the point cloud and $\sqrt{\lambda_2}$ as a characteristic width (see corresponding columns in Tables 2 and A1). More precisely, $\lambda_1$ and $\lambda_2$ are the variance of points projection on the first and second axes). $\lambda_3$ describes the points dispersion around the plane given by

eigen-vectors $e_1$ and $e_2$: we have $\lambda_3 = RMS$.

    For the plane fit to be robust, the $RMS$, i.e. $\lambda_3$, needs to be small in comparison to the length of the cloud. Following Fernández (2005), we use the indicator:

$$M = \ln(\lambda_1/\lambda_3)$$

    The higher $M$ is, the better the fit. To estimate the reliability of the plane orientation, we must assess the linearity of the point

cloud, as multiple planes can fit a linear distribution of points, with similar $RMS$. Fernández (2005) suggest the indicator $K$:

$$K = \frac{\ln(\lambda_1/\lambda_2)}{\ln(\lambda_2/\lambda_3)}$$

    The lower $K$ is, the better the reliability of the fit. Low values of $K$ are obtained if $\lambda_3 << \lambda_2$ (i.e. if the point cloud has a good degree of planarity) and/or if $\lambda_/\lambda_2$ is close to 1 (by construction, $\lambda_1/\lambda_2 \geq 1$). Fernández (2005) run multiple numerical tests to estimate threshold values for $M$ and $K$ corresponding to good fit and reliability. They suggest we must have $M \geq 4$ and

$K \leq 0.8$. In order to have a direct estimation of the uncertainty on dip and dip direction of the fitted plane, we use a bootstrap method. This is done by drawing randomly, with replacement, $N$ points among the initial $N$ points of the point cloud, and fitting a plane to the new point cloud. This procedure is repeated 100 times. The standard deviations of the resulting dip and dip directions are given in Table 2 and Table A1.



**Table A1.** Deposition horizons characterization. $\sigma$ is the standard deviation of estimated dip and dip directions. See Appendix A for characteristic lengths and widths computations, and for the derivation of $M$ and $K$. Following Fernández (2005), a good fit and reliability correspond to $M \geq 4$ and $K \leq 0.8$

| Mean Altitude (m) | dip (°) mean | dip (°) $\sigma$ | azimuth (°) mean | azimuth (°) $\sigma$ | RMS (m) | Characteristic length (m) | Characteristic width (m) | M | K |
|---|---|---|---|---|---|---|---|---|---|
| 723.54 | 19.06 | 0.27 | 269.49 | 0.27 | 2.29 | 64.57 | 15.24 | 6.68 | 0.76 |
| 749.86 | 23.01 | 0.21 | 258.46 | 0.13 | 0.80 | 40.91 | 7.51 | 7.88 | 0.76 |
| 764.67 | 13.22 | 0.25 | 242.55 | 4.14 | 0.84 | 25.62 | 2.70 | 6.84 | 1.92 |
| **766.42** | **23.90** | **0.33** | **252.23** | **0.39** | **1.81** | **65.73** | **34.32** | **7.18** | **0.22** |
| 792.50 | 17.00 | 0.29 | 262.14 | 0.78 | 0.84 | 36.03 | 7.21 | 7.51 | 0.75 |
| 795.30 | 14.92 | 1.14 | 267.31 | 5.88 | 0.73 | 23.75 | 1.36 | 6.96 | 4.64 |
| 813.17 | 9.82 | 0.11 | 291.63 | 1.42 | 0.40 | 31.46 | 3.44 | 8.72 | 1.03 |
| 821.96 | 15.27 | 0.17 | 263.26 | 0.92 | 0.27 | 22.41 | 2.17 | 8.82 | 1.12 |
| 832.81 | 6.98 | 0.23 | 219.50 | 3.42 | 0.33 | 13.32 | 2.46 | 7.38 | 0.84 |
| 843.32 | 10.13 | 0.13 | 232.06 | 1.01 | 0.31 | 20.87 | 3.91 | 8.41 | 0.66 |
| 847.60 | 5.85 | 0.08 | 291.39 | 5.59 | 0.23 | 15.82 | 1.39 | 8.49 | 1.34 |
| 854.92 | 5.43 | 0.20 | 252.21 | 1.41 | 0.47 | 32.83 | 5.25 | 8.50 | 0.76 |
| 864.34 | 10.83 | 0.28 | 257.05 | 1.33 | 0.49 | 23.70 | 3.60 | 7.78 | 0.94 |
| 875.21 | 23.33 | 1.19 | 210.64 | 1.02 | 0.39 | 8.99 | 1.79 | 6.25 | 1.07 |
| 915.00 | 14.29 | 1.71 | 267.02 | 2.01 | 0.32 | 10.40 | 0.76 | 6.96 | 3.00 |
| 932.54 | 17.87 | 0.88 | 229.60 | 0.87 | 0.21 | 9.65 | 1.18 | 7.66 | 1.22 |

*Author contributions.* Conceptualization, Methodology and Validation M.P., Y.L., A.N., T.D.; Formal Analysis, Investigation and Visualization M.P., Y.L., A.N., T.D.; Resources and Data Curation M.P., Y.L., A.N., T.D.; Writing Original Draft Preparation M.P.; Writing Review & Editing M.P., Y.L., A.N., Y.T, C.L., T.D., B.V.; Supervision, Project Administration, and Funding Acquisition A.N., B.V, Y.L., Y.T, C.L.

*Competing interests.* The authors declare that they have no conflict of interest.

*Acknowledgements.* We gratefully thank the DGPR and DEAL Martinique for funding this research and the acquisition of topographic data. We also thank the staff of BRGM Martinique, BRGM Guadeloupe and OVSM-IPGP for their contribution to field work.



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
