# Peer review of "How volcanic stratigraphy constrains headscarp collapse scenarios: the Samperre Cliff case study (Martinique Island, Lesser Antilles)"

_EGUsphere, 2022_

## Author Comment (AC1)

**We thank the reviewer for his comments. They are addressed below. Our answers are in bold.**

Dear authors,

Your work, although very interesting and with many methodological ideas for estimating slope instability in the absence of geomechanical/geotechnical data, is too speculative. That is, there are no counter-tests to your hypotheses. I find some main criticisms:

- identification of the "stable / unstable / less unstable" units: you have a profile with the different surfaces destabilized over time, you can use that to perform analyzes (for example Limit Equilibrium Methods, I recommend in 2D with Borselli's SSAP, https://www.ssap.eu/). As resistance parameters you can use those present in the literature (eg andesitic lavas, pumice, altered ashes, and so on) and carry out a series of tests by varying the parameters within the limits of the values found in the literature;

**In our view, a detailed limit equilibrium analysis with geotechnical model is beyond the scope of our study. We aim at providing a first hypothesis on the geometry of geological layers and their relative stability, using a geomorphological and descriptive approach. This hypothesis could indeed be tested in future works with limit equilibrium methods, including a sensitivity analysis on the geometry of interfaces, on the geotechnical properties of materials, and on the presence and levels of aquifers. Given the uncertainty on all these parameters, such a sensitivity analysis is, in our view, to complex to be included in the current manuscript. However we agree that future works could include limit equilibrium studies, and this will be added in the discussion.**

- palaeo-geographic reconstruction: since you are reconstructing the palaeo-geography of an entire flank of the volcano, I think that similar reconstructions were carried out by those who did geological mapping. Are your reconstructions consistent with those?

**We are not reconstructing the paleo-geography of an entire flank of the volcano, we are only focusing on small portion of the West flank of Montagne Pelée. To our knowledge, this has not been done before in the case of the Montagne Pelée. However, the succession of eruptive periods considered is in line with recent studies, e.g. Boudon and Balcone-Boissard (2021).**

- volume estimation: everything depends on point 1 (identification of the parameters). If you find the parameters, you could "test" your reconstruction by comparing it with some 3D instability models (for example using the parameters identified in 2D with SSAP within 3D models with SCOOPS-3D, https://www.usgs.gov/software/scoops3d). Since yours is a methodological draft, the proposed method should be tested with existing methods. Otherwise your results are too speculative.

**As stated before, such an analysis would require a detailed sensitivity analysis that we think is beyond the scope of this study. However, your suggestion will be added in the Discussion.**

Commentaires PDF

In realtà le fasi di allargamento dipendono sia dalle fasi di costruzione, ma soprattutto da quelle di distruzione dell'edificio. Ci sono vari studi sull'evoluzione morfologica degli edifici vulcanici che dimostrano che le fasi di smantellamento (catastrofico o progressivo) tendono a spostare il materiale vulcanoclastico sempre più a valle.

Consiglio di leggere:

De Rita, D., Giordano, G., & Milli, S. (1997). Forestepping-backstepping stacking pattern of volcaniclastic successions: Roccamonfina volcano, Italy. Journal of volcanology and Geothermal Research, 78(3-4), 267-288.

Manville, V., Németh, K., & Kano, K. (2009). Source to sink: a review of three decades of progress in the understanding of volcaniclastic processes, deposits, and hazards. Sedimentary Geology, 220(3-4), 136-161.

Di Traglia, F., Fornaciai, A., Favalli, M., Nolesini, T., & Casagli, N. (2020). Catching geomorphological response to volcanic activity on steep slope volcanoes using multi-platform remote sensing. Remote Sensing, 12(3), 438.

**We agree. This will be added in the Introduction, along with the provided references.**

*Regarding debris avalanches and debris flows* : in what sense are they associated? Is there a transformation of flow from rock avalanche to debris flow? Or is there the erosion and transport of the rock avalanches material by meteoric waters to form debris flows ?

**Once materials are deposited at the bottom of the cliff, they can be remobilized by water (immediately or after several weeks/months) to generate debris flows. This will be clarified.**

I cannot understand how you characterize the geological or geolithological unit

**The different geological units are identified by changes in colors and textures. Between the UPD and LPd units, and between the UPd and Co units, these changes are also associated to slope breaks.**

Sorry but it's not clear to me. If I understand correctly, you identify the "stable" units because they have not collapsed, while those that have been involved in the landslides that have already occurred are unstable. It's right? However, the stability / instability of a slope can depend on several factors, first of all the mechanical / technical characteristics of the material, and the slope. this is to say that a "stable" unit can become unstable if the morphology becomes leaning enough to cause it to collapse.

**We agree and will clarify this point. By « stable », we meant « more stable », given the current morphology of the cliff.**

Is the accuracy of 1 m on the z? of all the dem? or their difference? Please specify.

**The 1 m accuracy corresponds to the horizontal resolution of the DEM. This will be stated more explicitly.**

A propos du passage des horizons aux plans : It will certainly be my limitation, but I don't understand how you extrapolate the information on the plans.

**The horizons are 1D features, but the curvature of the cliff implies they are not lines. Thus, the corresponding points can be used to fit planes.**

I would not call them "stable", but "more stable than other units". I think it is more correct formally and it is a form that avoids misunderstandings.

**Agreed**

---

## Author Comment (AC2)

**'Comment on egusphere-2022-153', Anonymous Referee #2, 28 Jun 2022 reply**

**We thank the reviewer for his comments. They are addressed below. Our answers are in bold.**

The manuscript by Peruzzetto and co-authors focuses on the assessment of collapse scenarios on volcanic islands using an example from the Martinique Island. To achieve the study goals, observations from assessable stratigraphy are used to construct surfaces that can potentially act as rupture interfaces.

An aspect that can also be worth clarifying since the start of the manuscript is what is aimed at when mentioning reconstruction of the paleo-valley. I mention this just avoid early misinterpretations as quite often the term is used to recreate a pre-collapse morphology and calculate volumes evacuated from the slope. In this work, the "reconstruction" focuses more on the interpretation of a paleosurface buried by posterior volcanic deposits.

**Agreed, we'll make it clear that we are looking for a paleo-surface.**

The manuscript is well written and generally clear, apart from some occasional typos here and there. I made some suggestions on this, but a final proof-read is recommended. The figures are also of good quality and clear. However, and as commented also below, the colour codes used for the geological units in figure 3 do not correspond to the ones mentioned in the text, so this needs to be addressed. Please see below for my comments on this. Despite their number, I believe mostly are easy to address, and quite a few are just spot corrections.

**Title:**

The mention to 3D volcanic stratigraphy is good to draw attention, but after reading the content how much of 3d stratigraphy was actually able to be identified? There are some assumptions and manually created surfaces to create a hypothetical rupture surface, but not much beyond that. I'd suggest dropping the 3D from the title and keep the rest, as it will hold in relation to the content of the manuscript.

**Agreed**

**Introduction**

Line 20: in addition to these aspects, shouldn't vertical volcano-stratigraphic heterogeneity be mentioned as well? The alternation between different lithologies is itself a factor upon which all other will act upon, from just the gravity effect to fluid-related processes (derived from differences in permeability and flow properties). Although the accumulation of eruptive material is mentioned in the previous paragraph, the implications of distinct poro-perm properties is not.

**Agreed**

Line 36: not disagreeing with the statement that numerous small collapses may constitute a more immediate risk compared to large ones, how far can the statement of similarity between

them be supported? Numerous small collapses may be limited to the proximity of the edifice, and if far travel by these takes place the flows will be somewhat contained to streams and ephemeral flow paths, with immediate risk to the vicinity of these features. A major flank collapse may obliterate a lot on its flow path. Can this similarity stand?

We agree that large flank collapses have a larger impact. In this perspective, their study is indeed very important. However, the associated risks must take into account their temporal occurrence probability, which is smaller than for intermediate landslides as considered in this study. Such intermediate landslide may well occur far from current eruptive centres, in ancient volcanic formations no longer (or very rarely) affected by eruptions, and thus in urbanized areas. When they occur close to eruptive centers, they can still be a direct threat for populations when slopes are cultivated, or for hikers. Finally, the remobilization of their deposits by water can generate debris flows that are not necessarily contained to streams: this is the case for our case study, where debris flows sometimes flood the Prêcheur river, at the mouth of the river.

Line 51: on the mention that the determination of the landslide extent is a problem by itself: agree, but can a line of two be added on how this is a problem to complement the methods through which it is inferred? The first thing that usually comes to mind is the volume balance issue, as the calculated evacuated volume may not correspond to the inferred deposit. The deposit volume can be lower if part of the material is washed out and not trackable, or much larger as debris flows entrain more material. The latter is markedly significant for downslope risk assessment.

**The most direct way to estimate landslide volumes is indeed by measuring the volume of the associated deposits, although, as you point out, it not easy. However, this can be done for past landslides, not for potential landslides as there are no deposits yet. We can indeed add a line to explain this in the manuscript.**

Lines 63 to 65: very identical statements on the complexity and heterogeneity of volcanic complexes were made just some lines above. Please edit where more appropriate to avoid repeated content.

**We will make the changes at the beginning of the manuscript.**

Methods:

Line 85: based on what is described and referred to figure 1, the latter could include a map with the broader geographical location of the Lesser Antilles.

**Agreed**

Line 88: can the "first stage" please be clarified? Is it first eruptive stage? The geological succession itself is not strictly a stage as well, but the sequences produced by the volcanic activity. I suggest a slight rewording of these sentences. Please consider this comment on stage and the need to describe what it is (eruptive/active/volcanic/other as appropriate stage) valid for all instance ahead as well.

**It is indeed the first eruptive stage, this will be clarified.**

Line 110: as there was a change in paragraph, starting the sentence with "Such…" is not adequate. Either move this to the previous paragraph or add a few more words to clarify what type of avalanches the current paragraph will address.

**Agreed**

Line 129: seems to be a word missing. "... need to study/analyse/other? the geology of the.."

**Indeed, we'll add analyze**

Line 135: although mention to table 1 is made, please indicate just the number of additional sets acquired between 1988 and 2018.

**Agreed**

Line 136: a reference to figure 2 would be appropriate here.

**Agreed**

Line 145: is this 3D point cloud and the horizons it shows represented in any figure? If so, please cite it.

**It is visible in current Figure 5. It will be moved closer to the reference to the point cloud.**

Line 153: please indicate where the reader can see these polylines. Are these the limits of interpreted areas in figure 3?

**Agreed. They are visible in current Figures 3 and 5.**

Line 163: I'd suggest finishing the sentence with "... successive destabilization events/episodes".

**Agreed**

Line 164: "...units that are stable." Correct same typo in line 167.

**Agreed**

Line 168: "... have remained.."

**Agreed**

Line 188: please clarify the colour corresponding to unit La. The text mentions blue but the figure shows it as orange.

**Agreed, this is a mistake.**

Line 190: I suggest editing to "Most of the cliff below units Pu and La is..."

**Agreed**

Line 190: Unit UPd is shown in figure 3c as a pinkish colour. Clear purple is associated with unit LPd, which in line 209 is described as orange. Please correct all colour descriptions.

**Agreed**

Line 193: wouldn't it be clearer to just azimuth as N254 instead of N254E?

**Agreed**

Line 198: "dip angles.." "dip directions.."

**Agreed**

Page 9 in general: some observations on lithology and/or water seepages are made. Not doubting their presence or interpretation, are there any aspects in the figures that can help to support such observations?

**They can be seen in current Figure 4c. This will be added in the text**

Line 221: "...pyroclastic deposits..."

**Agreed**

Page 9 and 10: these bibliography-based interpretations of the rock units are adequate given the limitations of sample collection. Are there any further descriptive aspects taken from the pictures that can support some of them, given the good quality available. For instance, is anything else observable for unit Pu to support the interpretation of pumice, beyond its colour?

Further down the valley at the RPRE station (> 2km downstream of the Samperre cliff), we were able to sample and date pumice deposits that we interpret as another outcrop of the Pu unit. The radiocarbon age obtained ( $1285 \pm 25$  Cal AD, Nachbaur et al., 2019) is perfectly consistent with the age retained for the P1 eruption: 1300 AD (Carrazzo et al., 2012). We'll add this information in the manuscript.

Line 231: do the authors mean "were previously covered"? If this is to refer to multiple episodes, use re-covered instead or recovered as these imply distinct meanings.

**« Recovered » will be changed to « previously covered »**

Lines 230-235: a possibility to support the premise of hardened units C0 and LPd would be to show small topographic profiles across them. Being hardened, they would show as small bulges less prove to erosion and remobilisation than the adjacent UPd.

**We did not see such bulges.**

Line 236: This sentence needs to cite figure 7 for the reader to understand what is being talked about and see the surfaces. The sentence also needs to be re-written for clarity, namely in what regards the use "respectively X". An issue is that citing figure 7 for this will compromise the figure citation order as figure 6 has not been cited in the text so far.

**Agreed. We can invert Figures 6 and 7.**

Line 244: Was the post-collapse infill of the accommodation space by LPd associated with some sort of stratal dip, i.e., the LPd beds were not horizonal at deposition? If so, that is not captured neither by the written interpretation nor the diagrams in figure 6. Some clarification on such property could be useful. The same is valid for the stages where UPd deposited. Those clearly have evidence of dip, supported by the data, which could be represented in the diagrams (even if exaggerated for representativity).

The only clear deposition horizons that could be spotted are for the UPd unit. They have a slight dip of 16°, in the direction of the paleo-valley. As Figure 6 is a cross-section perpendicular to this paleo-valley, it is difficult to represent graphically this dip, but it can be indicated by text. We have no clear indication of deposition dip and orientation for the LPd unit. The only data is the dip angle and dip direction of the contact between LPd and UPd (S0 in Fig 6). It will also be displayed in Fig 6.

Line 245: this interpretation needs to be supported by figures for the reader to follow the process.

**References to current Figure 6 will be added.**

**Discussion:**

Lines 285 to 296: this paragraph discusses and supports quite well the interpretation of a stabler/indurated C0. It was one of the first questions that popped to my mind at the start of the discussion, and there may be a comment or other above on that, but this adequately addresses it using literature examples. However, these may still be considered somewhat speculative given the lack of sampling to support the ideas

**Agreed, this will be highlighted in the discussion**

Line 300: as we cannot be truly sure, at least not without samples, I suggest changing "certainly" for "likely" and keep interpretations open.

**Agreed**

Line 326. It is hard not to consider that water drainage, at surface and subsurface levels, is not be a key driver in erosion and shaping of the slope, be it at continuous sediment removal or more dramatic collapses. Is there any possibility of comparing rainfall data to morphological changes in the different periods of 2018, if relevant or applicable? If not driven by drainage, what other processes could be speculated for collapses on this setting?

As follow up, and despite the claim for further data to support or not the link of collapse to drainage, reading further ahead the authors dedicate a discussion point to the effect of groundwater circulation and links to precipitation as the main destabilisation mechanism. This somewhat contradicts what is stated or doubted in point 5.2. Even if miss-interpretations could derive from a possible ambiguous meaning of "drainage" in the paleo-valley as only surficial runout, the frequent mention to water seepages does seem to imply that groundwater flow is part of it. Please adjust the discussion to make it concise on this matter.

**By « drainage of the paleo-valley », we mean the progressive removal of the solid materials that fill the paleo-valley (i.e. materials from UPd unit). Paragraph 5.2 is about discussing if landslides contribute to this drainage. Water has indeed a role to play in this drainage, as evidenced by water seepages.**

Line 335- Point 5.3: I just want to add that this point seems, to me, well achieved. It mentions scenarios, impacts and uncertainties. Questions that could be raised for some discussed aspects were adequately clarified in following sentences.

**Thank you.**

Line 393, Referring to the whole paragraph: can averaged volume remobilisation per year be a reliable indicator to estimate evacuation trends? Collapses tend to be relatively "instantaneous" and frequencies are variable, so what is the risk of averaged rates leading to inadequate comparisons between different examples?

**We agree that these calculations should be taken with caution. The relative intense activity of the Samperre cliff over the past decades may lead to over-estimate the sediment production rate. This will be stated more clearly in the paragraph from 1.407 to 1. 415.**

Line 403: This comment could have been made before, but still relevant here: Brunet et al 2016 refers to a flank collapses in offshore settings, a scenario that is quite different from the one presented for the Samperre ravine. The volumes involved are also drastically different as submarine slides tend to be much larger, as the numbers here provided show. Despite the interpretation of multiple collapses on the flanks of the volcano, and some material derived from slope collapse, there is a large amount of material derived from basin sediment, so flank retreat is not directly comparable with the volumes provided as example. Referring back to the previous comment, how valid are averaged rates, especially for such long-spaced events within a time frame of 130k yr?

**The flank collapses discussed by Brunet et al 2016 resulted in offshore deposits, but they were initiated above the surface by massive destabilizations of previous volcanic edifices. As for the rates derived from the Samperre cliff, averaged rates are indeed uncertain. Thus, the comparison should only be made on the order of magnitude of the rates. This will be stated more clearly.**

Line 416 - on the examples of other erosive processes listed that may be dominant over edifice collapses: The retrogressive erosion can happen derived from different processes, so it is valid, although big collapses can still be the main driver for retrogression and sediment removal, with posterior morphological smoothing. On the other two examples mentioned, landslides (s.l.) were the main cause for strata removal, even if triggered by distinct processes.

How is the distinction in process dominance established, when based on the examples given the dismantling seems to mainly occur through the same one, i.e., slope collapse?

The manuscript will be clarified on this point. The distinction between erosive processes is indeed complex, because they follow one antoher. For instance, landslide deposits are remobilized by runoff. Salvany et al. (2012) suggest that regressive erosion by small landslides and runoff was enough to create the cirques in the Reunion Island, without massive flank collapses. Chaput (2013) acknowledges the role of regressive erosion by surface water and small landslides, but argue that they affect breccias emaplaced by massive flank collapses, the latter having initiated the formations of the cirques. Rault et al. (2022) highlight the important role of cyclone rainstorms in triggering slope failures, affecting both intact volcanic formations and epiclastic deposits. The relative importance of each process (runoff, major collapses and smaller landslides) is quantified by comparing sediment fluxes in rivers to catalogues of historical landslides and major flank collapses. The latter are identified from the geomorphological analysis of the current topography, and geological mapping of their deposits.

---

## Author Comment (AC3)

**'Comment on egusphere-2022-153', Georges Boudon, 08 Jul 2022 reply**

**We thank the reviewer for his comments. They are addressed below. Our answers are in bold.**

Review of the article egusphere-2022-153:

How 3d volcanic stratigraphy constrains headscarp collapsescentios : the Samperre Cliff case study (Martinique Island, Lesser Antilles)

by Marc Peruzzetto, Yoann Legendre, Aude Nachbaur, Thomas J.B. Dewez, Yannick Thiery, Clara Levy and Benoit Vittecoq

This article present how remote observations can be used to estimate the surface envelope of an unstable mass on a volcanic cliff. The case of the Sampere cliff on Montagne Pelée that show in the last decades a succession of detabilization was chosen.

The methodology is interesting to delimit an unstable zone in the absence of access to geological data on the ground.

The article is well written and well illustrated. The abstract and introduction present clearly the subject. There are a few typos. I encourage the authors to have a detailed re-reading of the manuscript.

I am not very competent to discuss the methodology used, based on the acquisition of photos and ortho photos, DEM... their compilation and modelling to estimate the geometry of the unstable area and the volume, but I can propose a number of criticisms on the geological data. But It seems to me, however, that the reconstruction and interpretations given are a little speculative and deserve to be a little more substantiated.

The nature of most of the deposits can be confirmed on the field. In the geological reconstruction of the different parts of the cliff, although access is difficult, I think that a study of the collapsed products at the foot of the cliff (while remaining in complete safety), or in the lower parts of the valley would help to better constrain the geological characterisation of the different formations that constitute the cliff. From what I know, I can note that a part of the collapsed products are made up of scoriaceous (low silica andesitic) products belonging to eruptions produced between 36 and 25 ka and corresponding to pyroclastic density currents (formerly called scoriaceous pyroclastic flows or locally St. Vincent type pyroclastic flows because they are comparable to those produced by the historical eruptions of the Soufrière of the island of St. Vincent in the southern part of the Antilles arc).

We agree, but unfortunately this is beyond the scope and funding of this study, whose objective is to propose a methodology to reconstruct the geometry of a paleo-valley using remote observations only. In the discussion and perspectives, we will add that deposits of lahars recovered in the Prêcheur river could be used as proxys to estimate the age, mineralogy and geochemistry of the UPd unit. These data could then be compared to samples of other sites to identify similar outcrops, where the same kind of data will have been acquired, and where the UPd unit is intact. Intact samples could then be extracted from these outcrops to determine their geotechnical properties. These products are topped by pumice fallout from the Plinian eruptions that have been occurring for 25 ka and probably few fine deposits from lava dome collapses (more abundant in this period).

**We agree. This corresponds to our unit Pu, but we will add in the manuscript that the top of unit UPd may also contain pumices fallout.**

I am a bit surprised by the presence of a lava flow in the upper part of the cliff (lines 187-190) for 2 reasons: i) in the past 36ka period of activity, few if any lava flows were emitted and in no case in this part of the volcano; ii) a lava flow, even if altered, would maintain a certain stability to the cliff and would form large blocks of lava when it collapsed, which are not found in any case in the collapsed products. Some pyroclastic density currents are welded on Montagne Pelée. It is for example the case of a pyroclastic density currents observed in a quarry near Ajoupa Bouillon on the southern flanc of the volcano. These deposits form prisms visible in this quarry. It is also the case of the « dalle soudée » located on the southern flanc of Etang Sec Crater. These deposits result from explosive eruptions that occurred in the last 25 000 years. The authors give the example of the lava flow of Morne Macouba dated at 12 ka. But this lava flow in not really a lava flow but a lava dome built on a slope that give a lava flow (it is a dome flow). No lava dome were produced in the last 25 000 years in this area. If it is really a lava flow, it indicated that all the deposits below belong to the period of activity 127-36 ka.

**We thank the reviewer for this correction in our interpretation of unit La. This will be corrected in the manuscript. As unit La is the top most unit of the cliff, this modification will not change our hypothesis on the paleo-morphology of the study site.**

Independently of this lava flow it is also probable that the deposits in the lower part of the cliff belong to this old period of activity1(27-36 ka) for example the unit LPd that is more resistant that the upper deposits attested also by the numerous cliffs observed on the southern flank of the volcano as for example the Tombeau des Caraïbes. It is what you propose in the article. It is quite possible that the Co surface could correspond to the floor of the flank-collapse structure formed 127 ka ago. This surface is a discontinuity that can correspond to a ''couche savon'' following the circulation of fluids and favors the instability of the deposits above. But the second phase of the volcano construction (127-36ka) is mainly made up of deposits of ash and blocks pyroclastic density currents associated with collapsed lava domes. These formations in the western part of the volcano are relatively indurated and have been cut by numerous radial valleys forming resistant cliffs that are not very prone to collapse (e.g. Tombeau des Caraïbes). It is therefore likely that the Sampere Valley cliffs correspond to a paleo-valley filled with post 36 ka products, which are less indurated and more prone to collapse.

**This is also our interpretation. In Figure 6, we assumed the lower part of the cliff, unit LPd, was emplaced between 127 ka and 36 kyrs, and eroded before the emplacement of the upper deposits UPd.**

In the discussion (lines 280-285), you state that it is not possible to characterise the geotechnical properties of the different units on the cliff because it is too dangerous which I understand. On the other hand, if you clearly identify the different units (as you have done for one part and can refine with my proposals), it is possible to acquire these data on deposits of

these different units in other parts of Mount Pelee. I can, if you wish, indicate perfectly accessible sites. These data would be very useful to support your hypotheses.

Unfortunately, the fees associated to the associated field mission and geotechnical tests are not compatible with the budget of our study (especially as at least 10 samples should be recovered and analysed to estimate a reasonable range of geotechnical properties). However, we agree that this could be done in future works, and this will be added in the discussion. Your suggestions about possible sampling locations are most welcome.

Lines 289-290: I am not sure that hydrothermal alteration at Co and the floor of the 127 ka flank-collapse structure stops after destabilisation. Instead, these discontinuities are zones of fluid flow (hot and cold), which continues the alteration and serves as a favorable « couche savon » for instabilities of the deposits above.

We agree that hydrothermal alteration of unit Co may have continued after the 127 kyrs from some time, at least at the interface, and will modify the manuscript accordingly. However, our interpretation is that fluids circulations (geothermal and/or meteorological) now occur at the base of Upd units, at the interfaces between LPd and UPd, and between Co and UPd. Observed destabilizations mostly affect UPd, but not Co. If Co was wtill being altered, it is destabilizations from this unit that would have triggered destabilizations from unit UPd, which does not seem to be the case.

Lines 298-299. I am not sure that all these deposits are welded by high temperature just after emplacement but probably indurated by compaction, diagnetical cementing...

**We agree to change the manuscript accordingly. However, could you provide some justifications on why high temperature welding does not seem likely ?**

5.2 : cliff destbabilizations... :It is obvious that a significant contrast exists between the lower formations (Co Lpd) and the upper formations (Upd) in terms of compaction, permeability, stability... Deposits from the post 36 ka eruptions fill many valleys on the western flank of Montagne Pelée cut into the Lpd formations and yet this type of destabilisation is not observed. We must therefore look for another explanation, such as a very steep contact between the two which could explain these destabilisations. As mentioned earlier, geophysical, electric, electromagnetic data (e.g. electrical soundings, such as those done at the Soufrière in Guadeloupe) could be collected in other Pelee valleys where this sequence of deposits is present

**We agree that the absence of other « Samperre cliff »-like catchments in Martinique can be surprising. Some reasons, that will be added explicitly in the manuscript, include :**

- The presence of the Prêcheur structure collapse on the Northern side of the Samperre cliff. The South part of the structure may have been affected by eruptions more recently than the Samperre cliff, and the deposition of associated volcanic materials compensated for the regressive erosion occurring in paleo-valleys.
- The destabilizations from the Samperre cliff have started no longer than 70s years ago, and may end in the coming decades (if our estimations of potential unstable volumes are correct, and if destabilizations keep occurring at the same rate). This is a rather short time scale in comparison to geological processes.

Thus, the evidence for similar processes that occurred in the past in other catchments may well have disappeared as a result of erosion by surface water and/or subsequent smaller gravitational re-adjustments.

- As you suggest the relative steep contact between basal units and UPd.
- The potential specificity of the hydrogeological watershed of the Prêcheur river.

We agree that comparisons with other sites could be done, and if you have suggestions of such sites, they are most welcome. If AEM data is available on these sites, we could use them in the discussion. A potential candidate could be the top of Rivière Chaude, on the South-West side of Montagne Pelée cone. Indeed, it displays scars of recent, and superficial, slope instabilities, and it is located near the Rivière Sèche collapse structure. But it is characterized by more recent volcanic materials from the 1902 and 1938 eruptions. As stated before, new acquisition campaigns are beyond the scope of this study.

5.3. Volume estimation : There are a number of unknowns about the geometry of the various contacts which leaves the estimation of volumes very approximate, but it does give a rough idea of the volumes that could destabilise. As a result, the scenarios remain highly speculative.

**We agree, this is why we considered different scenarios. As you say, the volume estimations should only be considered as « rough » estimations. However, they do give an idea of what could still be remobilized, and compare it to the volume that has collapsed over the past 20 years.**

The approach of the article is interesting, the methodology as well, but a number of improvements are necessary in the identification of the depositional sequences but also in the improvement of the geometry of the different units which in my opinion can only be done by the acquisition of geophysical and geotechnical data on other valleys of the Montagne Pelée that can serve as an analogue to the Sampère cliff. Otherwise the results remain very speculative. I therefore consider that the approach is interesting, that this type of data is necessary, but that the article can only be published after significant modifications.

**As stated above, the budget associated to these new acquisitions is beyond the scope of this paper. But we will state explicitly in the manuscript that it should be done in future works.**

---

## Author Response (AR2)

**Final modifications, EGUSPHERE-2022-153**

**How volcanic stratigraphy constrains headscarp collapse scenarios : the Samperre Cliff case study (Martinique Island, Lesser Antilles)**

**Peruzzetto et al.**

We thank the editor and reviewer for accepting the manuscript for publication. The only modification to the manuscript in comparison to the previous version is in the acknowledgments. We now thank the editor and reviewers.